# Blood Vessel Patterning on Retinal Astrocytes Requires Endothelial Flt-1 (VEGFR-1)

**DOI:** 10.3390/jdb7030018

**Published:** 2019-09-07

**Authors:** John C. Chappell, Jordan Darden, Laura Beth Payne, Kathryn Fink, Victoria L. Bautch

**Affiliations:** 1Center for Heart and Reparative Medicine Research, Fralin Biomedical Research Institute, Roanoke, VA 24016, USA; 2Department of Biomedical Engineering and Mechanics, Virginia Polytechnic Institute and State University, Blacksburg, VA 24061, USA; 3Department of Biology, The University of North Carolina at Chapel Hill, Chapel Hill, NC 27599, USA; 4Graduate Program in Translational Biology, Medicine, and Health, Virginia Tech, Blacksburg, VA 24061, USA; 5McAllister Heart Institute, University of North Carolina at Chapel Hill, Chapel Hill, NC 27599, USA

**Keywords:** *flt-1*, VEGF-A, angiogenesis, retina, blood vessel development

## Abstract

Feedback mechanisms are critical components of many pro-angiogenic signaling pathways that keep vessel growth within a functional range. The Vascular Endothelial Growth Factor-A (VEGF-A) pathway utilizes the decoy VEGF-A receptor Flt-1 to provide negative feedback regulation of VEGF-A signaling. In this study, we investigated how the genetic loss of *flt-1* differentially affects the branching complexity of vascular networks in tissues despite similar effects on endothelial sprouting. We selectively ablated *flt-1* in the post-natal retina and found that maximum induction of *flt-1* loss resulted in alterations in endothelial sprouting and filopodial extension, ultimately yielding hyper-branched networks in the absence of changes in retinal astrocyte architecture. The mosaic deletion of *flt-1* revealed that sprouting endothelial cells flanked by *flt-1^−^*^/*−*^ regions of vasculature more extensively associated with underlying astrocytes and exhibited aberrant sprouting, independent of the tip cell genotype. Overall, our data support a model in which tissue patterning features, such as retinal astrocytes, integrate with *flt-1*-regulated angiogenic molecular and cellular mechanisms to yield optimal vessel patterning for a given tissue.

## 1. Introduction

Sufficient blood flow is vital for tissue and organ homeostasis. Blood vessel growth and remodeling must therefore be tightly regulated to achieve appropriate structural patterns and densities. Underdeveloped or malformed vascular networks lead to insufficient blood flow and thus, inadequate nutrient delivery [1,2], while vessel overgrowth is equally detrimental and also results in poor tissue oxygenation [3,4,5,6]. Negative feedback mechanisms within pro-angiogenic signaling pathways are critical for maintaining developing vessel networks within a range of sufficient but not excessive growth and for ensuring that vascular networks form in the appropriate locations. For example, endothelial cell responses to Vascular Endothelial Growth Factor-A (VEGF-A), among other angiogenic cues [2,7], require precise coordination, achieved, in part, through feedback regulation of VEGF-A receptor expression and activity [8,9,10]. In particular, the VEGF-A activation of endothelial cells can be modulated via a decreased expression of the pro-angiogenic VEGF-A receptor Flk-1 (VEGF Receptor-2) [11,12] and by an increased endothelial cell expression of a second VEGF-A receptor, Flt-1 (VEGF Receptor-1) [12,13,14,15]. Flt-1 binds VEGF-A with a 10-fold higher affinity than Flk-1, indicating that it can function as a ligand sink [16,17,18,19]. Impaired crosstalk between the VEGF-A pathway and other signaling cascades, such as the Notch pathway, can disrupt this critical feedback mechanism and lead to blood vessel remodeling outside the range of productive vascular growth [8].

Further highlighting the importance of Flt-1 in maintaining proper VEGF-A signaling is the observation that global *flt-1* deletion causes lethality at embryonic day 8.5–9 (E8.5–9), owing to vascular defects in the yolk sac and the embryo proper [20]. Flt-1 mRNA is alternatively spliced to give membrane-localized (mFlt-1) and soluble (sFlt-1) isoforms [21,22]. Both endothelial-derived Flt-1 isoforms are competitive inhibitors of Flk-1-mediated signaling and negatively modulate VEGF-A pathway activity during developmental blood vessel formation [23] and deletion of the Flt-1 intracellular tyrosine kinase domain is compatible with normal vascular development [24]. We and others have shown that Flt-1 has unique and important roles in coordinating endothelial sprouting dynamics [21,25,26,27] and blood vessel anastomosis [28], with a genetic loss of *flt-1* leading to vascular overgrowth, dysmorphogenesis [29,30], and an overall reduced network complexity [8,20,21,26,31]. The exact nature of the Flt-1 regulation of blood vessel formation seems variable and context-dependent, however, and inducible loss of *flt-1* in postnatal mice causes an abnormal increase in blood vessel branching in various tissues and organs, due in part to enhanced production and activation of Flk-1 [32]. In this study, we extended those studies to ask how Flt-1-regulated cellular mechanisms governing endothelial cell sprouting and vessel patterning are disrupted in the in vivo context of the developing mouse retina.

During angiogenesis, endothelial “tip” cells emerge from established blood vessels to lead the extension of nascent vessel sprouts. Flt-1, and soluble Flt-1, in particular, shapes the local gradient of available VEGF-A and contributes to the spatial guidance of tip cells [25,33]. Although filopodial extensions from these “pathfinding” cells are not required for vessel formation [34], directional migration of endothelial cells and proper spatial patterning of new vasculature is more efficient when filopodia facilitate the detection of ligand concentration gradients—specifically VEGF-A gradients [35,36,37,38]. In the current study, we asked how Flt-1 regulates blood vessel formation in a non-cell autonomous manner by refining the near-field gradients of available VEGF-A in a structurally defined environment. Specifically, we explored this relationship in the developing mouse retina, where retinal astrocytes provide a scaffold for VEGF-A presentation of endothelial cell filopodia, which effectively organizes vessel patterning in this organ [35,36,37,38]. Our observations show that Flt-1 regulated the extent of vascular growth and promoted the efficiency of endothelial tip cells and their filopodia in establishing new vessels in spatially defined locations along the astrocyte “template.”

## 2. Materials and Methods

### 2.1. Mouse Husbandry

As described previously [28], mice (*Mus musculus*) with the *Tg (UBC^Cre-ERT^*^2^*)* gene (Jackson Laboratory #007001) were bred with 2 additional mouse lines: (1) one line harboring a reporter gene for Cre recombinase (Cre)-mediated recombination, specifically *R26R^TdTom/TdTom^* [*Gt(ROSA)26Sor^tm14(CAG-tdTomato)Hze^*, Jackson Laboratory #007914], and (2) a line with *loxP* sites flanking the first exon of the *flt-1* gene [*Flt-1^loxP/loxP^* (from Napoleone Ferrara, University of California, San Diego, CA, USA; formerly at Genentech)] [25,28,29]. Ethical standards for animal use according to the University of North Carolina, Chapel Hill, Institutional Animal Care and Use Committee were maintained. 

### 2.2. Inducible Cre Excision and Retina Tissue Processing

Cre-mediated gene excision was achieved by administering 100 μg of tamoxifen (MP Biomedicals) in 10% pure ethanol and 90% sunflower seed oil (Sigma) via daily intraperitoneal (IP) injections from postnatal day 2 (P2) to 4. Using a similar protocol, we induced mosaic Cre-mediated recombination by administering 5 μg of tamoxifen at P2. Postnatal day 6 and P21 eyes were perfusion-fixed with 0.5% paraformaldehyde (PFA) in PBS, collected, and immersed in 2% PFA for 2 h at room temperature (RT). Following PBS rinse, the retinal layer was isolated from fixed eyes by micro-dissection and immersed in cold pure ethanol for 30 min. Retinal tissues were re-hydrated and permeabilized for 30 min in PBS-T (PBS + 1% Triton-X (Fisher)) at RT. 

Blood vessels were labeled by incubating retinas in isolectinB4 conjugated to AlexaFluor 488 (1:100, Invitrogen) and cell nuclei were labeled by DAPI (1:1000). For retinal astrocyte immunostaining, P6 mosaic retinas were incubated with 5% normal goat serum (Jackson ImmunoResearch) in PBS-T for 1h at RT, rat anti-glial fibrillary acidic protein (GFAP) (1:200, ThermoFisher) in PBS-T overnight at 4 °C, and goat anti-rat AlexaFluor 633 (ThermoFisher) in PBS-T for 3 h at RT. Following PBS washes, retinas were mounted in PBS:glycerol (1:1) and imaged using a Leica DMI 6000B or Zeiss LSM 880 confocal microscope at 40× or 63× magnification, with post-acquisition *z*-stacks compression using ImageJ software.

### 2.3. Quantitative Analysis of Retinal Vasculature

Retinas with Cre-mediated excision of *flt-1* were analyzed using ImageJ for vessel branch point density (branch points per vessel length, P6 and P21), percentage of vessel segments less than 30 microns in length (P6 only), and for vascular area per field of view (P6 only). For P21 retinas, each retinal layer (i.e., retinal ganglion cell (RGC) layer, inner plexiform layer (IPL), and the outer plexiform layer (OPL)) was assessed independently for branch point density, as done previously [32]. Applying previously established methods [25,35], the sprouting vascular front of P6 retinas was analyzed for the density of endothelial sprouts and their filopodial extensions (i.e., per vessel length of the vascular front), as well as for the frequency of bifurcated sprouts.

Conditional *flt-1* deletion and littermate retinas with Cre-mediated excision of *flt-1* were analyzed for GFAP+ astrocyte area for a given region of interest, as well as the percent overlap between GFAP+ astrocytes and isolectinB4-labeled blood vessels. In ImageJ, threshold cutoffs were applied to confocal images of both the GFAP and isolectinB4 signals to reduce background noise and create binary images. The total area of GFAP+ astrocytes within a given region of interest was found and then the “multiply” function was applied to both images to yield an image of pixels positive for both signals, thus facilitating the measurement of GFAP+—isolectinB4+ area. Percent overlap was then calculated by dividing GFAP+—the isolectinB4+ area by the total GFAP+ area. 

By observing TdTomato reporter expression in mosaic retinas, endothelial cells were identified as having undergone Cre-mediated genetic excision and thus, lacking a functional *flt-1* gene locus (isolectinB4^+^/DAPI^+^/TdTomato^+^: *flt-1^−^*^/*−*^ endothelial cell), or devoid of Cre activity and therefore competent for *flt-1* expression (isolectionB4^+^/DAPI^+^/TdTomato^−^: WT endothelial cell). Fine-grain mosaic areas were classified based on the excision status (i.e., TdTomato expression) of the endothelial cells within the sprout and within the vessel regions flanking the emerging vessel. These areas were then scored for the percentage of GFAP^+^ astrocyte “paths” immediately adjacent to the sprout that were occupied by isolectinB4^+^ filopodial extensions emerging from the endothelial sprout. The filopodia number per sprout and the filopodia angle of extension relative to the sprout main axis were measured. The percentage of endothelial sprouts with a bifurcated phenotype was also determined.

### 2.4. Statistical Analysis

A statistical analysis was performed with GraphPad Prism 6. Multiple measurements were made for each parameter. Where appropriate, averages and standard errors about the mean were calculated from these values. For the averaged measurements, statistical comparisons were made using an unpaired two-tailed Student’s *t*-test, except for the mosaic retina analysis, which required testing by ordinary one-way ANOVA, followed by Tukey’s multiple comparison test. Measurements that yielded percentages were compared by Fisher’s exact test or a Chi-square analysis where appropriate. The statistical analysis for each measurement is described in the corresponding Figure Legend. 

## 3. Results

### 3.1. Induced Genetic Deletion of Flt-1 Leads to Sustained and Excessive Vascular Growth In Vivo 

Proper blood vessel formation requires tight coordination of VEGF-A signaling, and the VEGF receptor Flt-1 is a critical element in the feedback regulation of this potent signaling cascade. We determined how Flt-1 regulates angiogenesis by conditionally deleting *flt-1* in vivo, shortly after birth, in a well-characterized vascular bed—the post-natal retina. Retinal vascularization occurs initially by radial expansion of a vessel plexus from the optical nerve area, then after about 8 days, the vessels sprout vertically to form two other layers, leading to three distinct vessel layers by P21 [37]. Specifically, we used mice in which the first exon of *flt-1* was flanked by *loxP* sites [25,29], and also carrying a TdTomato reporter gene for Cre activity downstream of a stop cassette flanked by *loxP* sites in the *ROSA* locus (*R26R^TdTom/TdTom^;*
*flt-1^loxP/loxP^*). These mice were bred to *Tg* (*UBC^Cre-ERT2^*) mice to facilitate the tamoxifen-induced deletion of floxed genes. The visualization of the TdTomato fluorescent reporter showed that Cre-recombinase activity was primarily, if not exclusively, confined to the endothelial cell compartment of P6 retinas and that the majority of retinal endothelial cells expressed the excision reporter (Appendix A). The morphological analysis of the P6 retinal vessels revealed that vessel networks with genetic deletion of *flt-1* had increased vessel branch points and vascular area compared to littermate controls, with a notable increase in vessel segments less than 30 microns in length (Figure 1a–e). The increased vessel density in the mutant retinas persisted throughout the course of eye maturation to P21 in all three vessel layers (Figure 1f–l), including the retinal ganglion cell (RGC) layer, as found in the same layer of P6 retinas. We found a similar trend of excessive filopodia following endothelial cell-selective *flt-1* deletion in embryonic back skin (Appendix A). These data support the idea that the loss of *flt-1* in endothelial cells disrupts mechanisms that regulate the proper density and patterning of blood vessels, as seen in the developing retina and in less stereotyped tissue beds such as the skin. Moreover, analysis of retinal Flt-1 expression via the *lacZ* reporter showed little to no apparent expression by astrocytes or by vascular mural cells at the sprouting vascular front of P8 retinas (Appendix A), suggesting that the primary source of retinal Flt-1 at this time-point in retinal development was endothelial cells. These observations are consistent with several published reports of heterogeneous but restricted *flt-1* expression in the retinal endothelium [39,40,41,42,43,44]. These data also align with our previous analysis of both *flt-1* gene expression and β-galactosidase antibody staining (i.e., *flt-1^lacZ/+^*), showing little to no detectable *lacZ* expression in non-endothelial vascular cells (specifically in NG2+ pericytes) in ES cell-derived vessels [45]. Therefore, based on (i) Cre-recombinase activity occurring largely in the retinal endothelium in our model (Appendix A), (ii) the similarity of phenotypes of endothelial-selective (Appendix A) and UBC-CreER-mediated excision of *flt-1,* and (iii) the restricted expression of Flt-1 in the retina endothelial cells [39,40,41,42,43,44], we used the *Tg(UBC^Cre-ERT2^)* driver for the remainder of the study, due to an improved experimental control over excision levels. 

### 3.2. Conditional Flt-1 Ablation Disrupts Endothelial Sprouting in the Developing Retinal Vasculature.

We then asked which aspects of endothelial cell behavior were affected by the conditional loss of *flt-1*, particularly with respect to endothelial cell sprouting migration dynamics. Conditional *flt-1* deletion increased the density of sprouting endothelial cells at the leading edge of the developing retinal vasculature (Figure 2a–c). Interestingly, we also observed an increase in filopodia and in endothelial sprouts that were bifurcated at the leading front of the conditional *flt-1^−^*^/*−*^ vasculature (Figure 2a,b,d,e). These phenotypes were recapitulated in endothelial cell sprouts of *flt-1^−^*^/*−*^ embryonic stem cell-derived vessels (Appendix A). These data indicate that disrupted endothelial cell sprouting dynamics contributed significantly to the vascular overgrowth observed in conditional *flt-1*^−/−^ mutant retinal vessels. 

### 3.3. Retinal Astrocyte Patterning Is Unaffected by the Conditional Loss of Flt-1

Since the patterning of developing retinal blood vessels is closely associated with the spatial distribution and architecture of retinal astrocytes, we asked whether conditional loss of *flt-1* alters the astrocyte template. We immunostained astrocytes in inducible *flt-1^−^*^/*−*^ retinas and littermate controls and found that the overall density and spatial distribution of glial fibrillary acidic protein (GFAP)+ astrocytes were comparable between the two groups (Figure 3a–g). Agreeing with previously published observations [35,36,37,38], the blood vessel pattern strongly coincided with the underlying astrocyte pattern in both groups. Although the retinal vasculature was not present on all the astrocyte “paths”, in any case, the maximal induction of *flt-1* genetic loss caused a significant increase in the percentage of the astrocyte network occupied by blood vessels compared to littermate controls (Figure 3h). Thus, conditional *flt-1* deletion had no detectable effect on the development and spatial patterning of retinal astrocytes, but it increased the extent to which retinal blood vessels occupied the underlying astrocytic network.

### 3.4. Mosaic flt-1 Expression at the Vascular Front Increases Endothelial Filopodial Extensions on Underlying Astrocytes

VEGF-A is produced in the developing retina by hypoxic retinal ganglion cells and astrocytes [46,47], as well as several other cell types [44,48]. Although astrocyte-derived VEGF-A production is not required for normal vessel patterning, it is required for endothelial cell survival [47]. These and other studies suggest that retinal astrocytes function to present VEGF-A to migrating endothelial cells. In addition, we previously demonstrated that Flt-1 activity during retinal vessel formation promoted the spatial guidance of emerging endothelial tip cells [25]. We therefore hypothesized that Flt-1 modulated astrocyte presentation of VEGF-A and blunted endothelial cell sprouting migration along the astrocyte template. To test this hypothesis, we used *Tg* (*UBC^Cre-ERT2^*); *R26R^TdTom/TdTom^;*
*flt-1^loxP/loxP^* mice and applied lower doses of tamoxifen to generate mosaic *flt-1* excision in developing mouse retinal vessels, confirmed by discontinuities in TdTomato expression (Figure 4a–h and Appendix A). In regions where *flt-1* was absent from one or both lateral base endothelial cells flanking emerging sprouts, these sprouts, independent of *flt-1* excision status, were more likely to extend filopodial projections along a wider range of astrocyte “paths” compared to those in WT areas (Figure 4i,j). Filopodial extensions were also more numerous from the WT and conditional *flt-1^−^*^/*−*^ sprouts when both lateral base cells lost *flt-1* (Figure 4k) and bifurcated sprouts were more frequent in this scenario as well (Figure 4l). Additionally, when both flanking endothelial cells were recombined and presumably lost *flt-1*, the angle at which a filopodium extended from an endothelial sprout was larger relative to the sprout main axis (Figure 4m), suggesting that Flt-1 expression in lateral base cells affected vascular sprouting in vivo. Taken together, these observations support the idea that endothelial cell-derived Flt-1 provides non-cell autonomous regulation of sprouting dynamics by modifying the presentation of available VEGF-A along the underlying retinal astrocyte network.

## 4. Discussion

For potent pro-angiogenic cues, such as VEGF-A, negative feedback regulation is critical for maintaining vessel growth within a productive range; how these feedback mechanisms integrate into overall blood vessel formation, however, is not well-understood. In this study, using conditional genetic recombination tools to manipulate the expression of the decoy VEGF-A receptor Flt-1 in developing mouse retina, we analyzed how Flt-1 can similarly regulate endothelial cell sprouting behaviors non cell-autonomously, as previously described [8,20,21,25,26,28,29,31], yet elicit differential vessel patterning outcomes [32]. Endothelial cells produce Flt-1, and this expression is important in establishing tip/stalk cell phenotypes [8,9,10] and to spatially refine the availability of local VEGF-A [25]. In vascular beds that form in the absence of an underlying pattern, such as the yolk sac [49] or ES cell-derived vessels, mis-regulation of VEGF-A signaling [26] leads to an overall decrease in network branching complexity, as seen in the reduced number of branch points and vessel segments [8,20,21,26,31]. Consistently with previous reports [30,32,50], we showed that the disruption of Flt-1 activity in the context of a spatially stereotyped vascular bed increased the overall branching, although similar endothelial sprouting and filopodia navigation mechanisms were affected by the loss of *flt-1*. Moreover, in both scenarios, the primary effects of *flt-1* loss appeared to be non-cell autonomous and likely depend on lateral base endothelial cells. Thus, Flt-1 regulation of VEGF-A signaling is an important negative feedback mechanism whose ultimate effect on vascular morphogenesis is context-dependent and it works in conjunction with other patterning cues, such as local tissue architecture.

The developing mouse retina has been widely used as a model for angiogenic growth and expansion. This vascular bed forms in the context of non-endothelial cells, such as neurons, maturing retinal ganglion cells, and astrocytes, which provide key molecular and structural cues for the developing vasculature. In particular, retinal astrocytes secrete high levels of VEGF-A, as do retinal ganglion cells, to promote new vessel formation and alleviate tissue hypoxia [46,47]. Astrocyte VEGF-A is dispensable for retinal angiogenesis [47], but astrocytes may provide structural “paths” for endothelial cells to use as a template as the vascular network expands [38,51]. We found that vascular sprouts did not cover all astrocytes, but rather, the endothelial cells appeared to “choose” a path forward from among several options. This process was affected by the loss of *flt-1*, whereby multiple paths were used to produce increased vessel growth and branching that was nevertheless spatially restricted to the astrocytic network. Since astrocyte maturation and vessel development are closely linked [51], we verified that the interplay between these two systems was not disrupted by conditional *flt-1* deletion and that the vascular defects observed were not secondary to changes in the astrocyte template. 

Endothelial phenotypic heterogeneity is essential for successful blood vessel growth and remodeling. The molecular mechanisms that establish and maintain such heterogeneity are still being clarified, but it is clear that elements of the VEGF-A [8], Notch [10], and bone morphogenetic protein (BMP) signaling pathways [7,52], among others, are critical mechanistic determinants of this heterogeneity. Heterogeneity in *flt-1* expression has been described in several studies as a key component of successful angiogenic remodeling [25,42,53]. In this research, we imposed mosaic expression of Flt-1 in endothelial cell sprouts and lateral base areas via genetic excision and observed how this loss of Flt-1 affected both cell autonomous and non-cell autonomous patterning. This unique approach provides insight into the cellular basis for the hyper-branched vasculature that we, and others [30,32,50] have observed in conditional *flt-1^−^*^/*−*^ retinas, even at later time points in eye maturation. Specifically, in this study, we showed that there was a non-cell autonomous role for Flt-1 expression, likely from lateral base cells, in shaping the forward trajectory of vascular sprouts in vivo, confirming and extending our previous results in vitro. Remarkably, the sprout phenotypes were independent of the sprout genotype, suggesting that the relevant isoform for this crosstalk is sFlt-1. This research also indicates that communication among neighboring endothelial cells via Flt-1 levels is a hallmark of Flt-1 regulation of vascular morphogenesis. This communication is essential for sprout guidance, as shown here and previously [8,25]. Interestingly, Flt-1 also appears important for additional phases of vascular network formation. For instance, proper vessel anastomosis and stabilization also utilize Flt-1 regulation of VEGF-A [26], with the membrane-localized Flt-1 isoform playing unique roles within these processes [28]. Thus, both cell autonomous and non-cell-autonomous negative modulation of VEGF-A signaling via Flt-1 appeared to integrate blood vessel patterning and stabilization.

Negative feedback mechanisms for robust pro-angiogenic stimuli are essential to regulate vascular growth and remodeling. VEGF-A is one of the most potent inducers of blood vessel remodeling and is implicated in a wide range of pathological conditions where mis-regulated vessel growth contributes to disease onset and progression. In this study, we provided new insight into one such feedback mechanism for VEGF-A signaling. The negative VEGF-A modulator Flt-1 provided key limitations on vessel remodeling in a given context by shaping the local signaling environment and affecting how endothelial cells integrated this information with other patterning cues, such as the architectural elements within a tissue. This increased insight will influence the design of therapeutic strategies and potentially mitigate the adverse side of VEGF-A therapy [54].

## Figures and Tables

**Figure 1 jdb-07-00018-f001:**
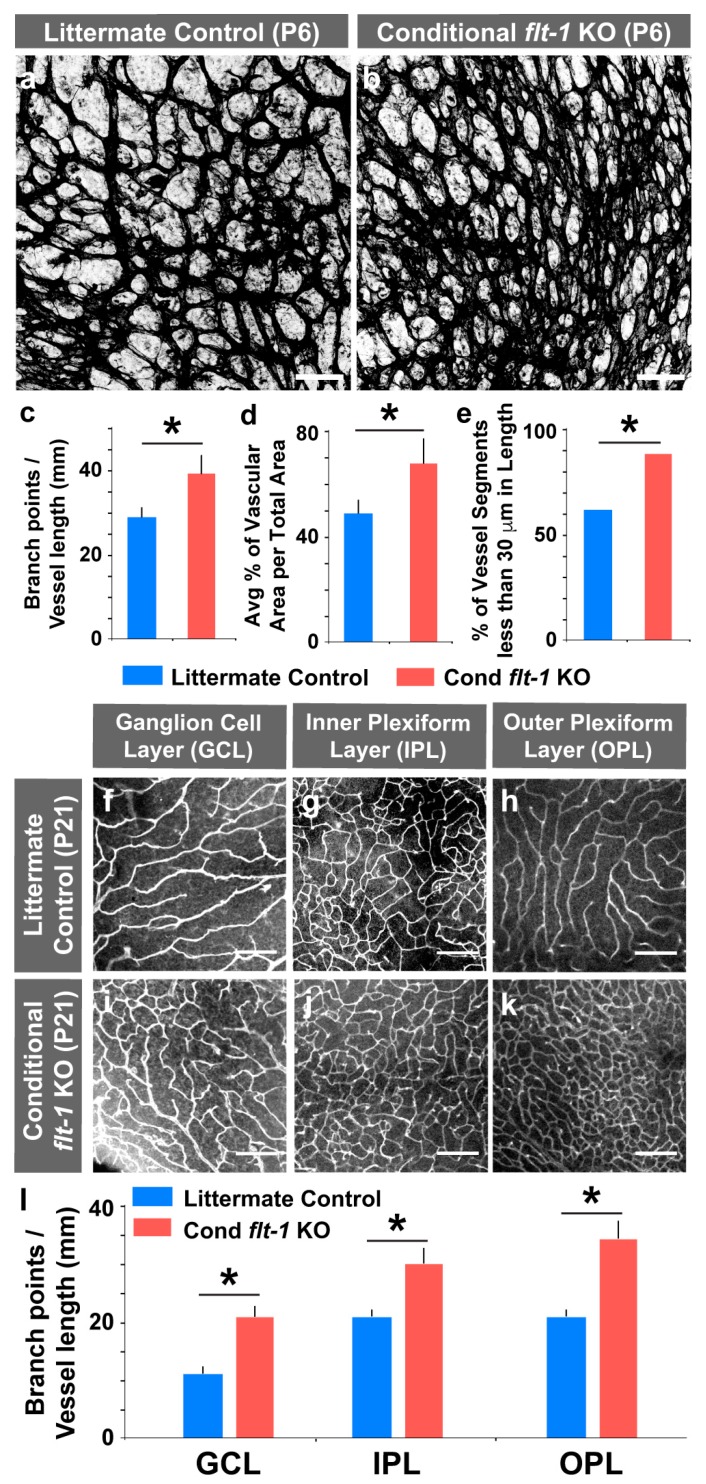
Conditional loss of *flt-1* increases the blood vessel density of postnatal mouse retinas. (**a**,**b**) Representative images of postnatal day 6 (P6) littermate control (**a**) and conditional *flt-1* knockout (KO) (**b**) mouse retinal vasculature labeled with isolectinB4 conjugated to AlexaFluor488. Scale bar, 50 μm. (**c**–**e**) Indicated quantifications of P6 retinal vasculature. (**c**) Retinal blood vessel branch points per total vessel length (mm) for littermate control (blue bars, *n* = 6) and conditional *flt-1* loss (Cond *flt-1* KO, red bars, *n* = 6) mice. The values are averages and the error bars represent standard error about the mean. * *p* < 0.05 by unpaired student’s two-tailed *t*-test. (**d**) Retinal blood vessel area per total area within a randomly selected region of interest for littermate control (*n* = 6) and conditional *flt-1* loss (*n* = 6) mice. The values are averages, and the error bars represent standard error. * *p* < 0.05 by unpaired two-tailed *t*-test. (**e**) Percent of vessel segments with lengths less than 30 microns. Values are percentages. * *p* < 0.05 by Fisher’s exact test. (**f**–**k**) Representative images of P21 littermate control (**f**–**h**) and conditional *flt-1* KO (**i**–**k**) mouse retinal vasculature labeled with isolectinB4 conjugated to AlexaFluor488 and acquired from each vascularized layer: retinal ganglion cell layer (GCL), inner plexiform layer (IPL), and outer plexiform layer (OPL). Scale bar, 100 μm. (**l**) Retinal blood vessel branch points per total vessel length (mm) for P21 littermate control (*n* = 6) and conditional *flt-1* KO (*n* = 6) mice measured from each distinct vascularized layer. The values are averages and the error bars represent standard error. * *p* < 0.05 by unpaired two-tailed *t*-test for each retinal layer.

**Figure 2 jdb-07-00018-f002:**
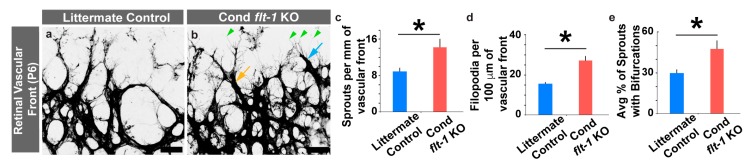
Conditional *flt-1* deletion disrupts endothelial tip cell sprouting dynamics at the vascular front. (**a**,**b**) Representative images of postnatal day 6 (P6) littermate control (**a**) and conditional *flt-1* knockout (KO) (**b**) mouse retinal vascular front stained with isolectinB4 conjugated to AlexaFluor488. The light blue arrow denotes an endothelial tip cell, the green arrowheads denote filopodial extensions, and the orange arrow denotes an example of a bifurcated sprout. Scale bar: 50 μm. (**c**–**e**) Indicated quantifications of vascular front of P6 retinas. (**c**) Endothelial cell filopodia per 100 microns of vessel length for littermate control (blue bars, *n* = 6) and conditional *flt-1* loss (red bars, *n* = 6) mice. (**d**) Sprouting endothelial cells per mm of vascular front for littermate control and conditional *flt-1* loss mice. (**e**) Average percentage of endothelial cell sprouts with bifurcated extensions for littermate control and conditional *flt-1* loss mice. The values are averages and the error bars represent standard error. * *p* < 0.05 by unpaired two-tailed *t*-test.

**Figure 3 jdb-07-00018-f003:**
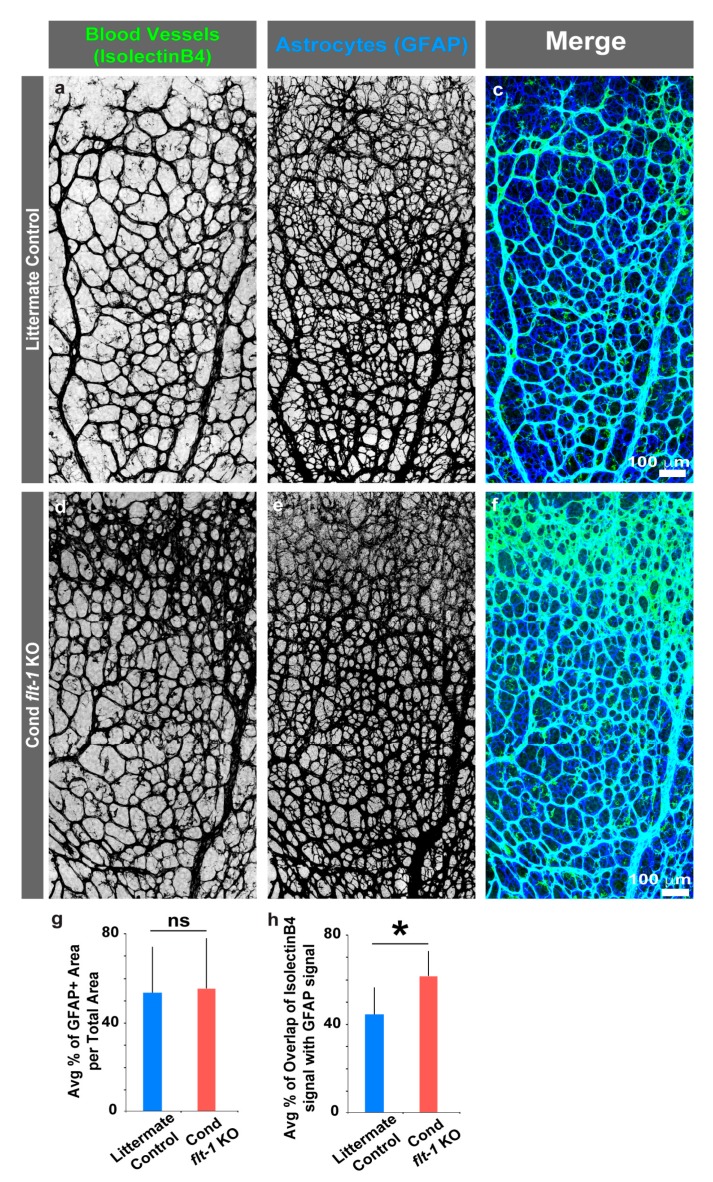
Blood vessel association with retinal astrocytes increases with conditional deletion of *flt-1*. (**a**–**f**) Representative images of postnatal day 6 (P6) littermate control (**a**–**c**) and conditional *flt-1* knockout (KO) (**d**–**f**) mouse retinas labeled for blood vessels (isolectinB4-AlexaFluor488 in **a** and **d**, and green in merged images **c** and **f**) and for astrocytes (glial-fibrillary acidic protein (GFAP)-Alexa Fluor633 secondary in b and e, and blue in merged images **c** and **f**). Scale bar, 100 μm. (**g**–**h**) Indicated quantifications. (**g**) Average percentage of GFAP+ area per total region of interest area for littermate control (*n* = 6) and conditional *flt-1* loss (*n* = 6) mice. (**h**) Average percentage of overlap of IsolectinB4+ signal with GFAP+ signal within regions-of-interest for littermate control and conditional *flt-1* loss mice. The values are averages and the error bars represent standard error. * *p* < 0.05; ns, not significant by unpaired two-tailed *t*-test.

**Figure 4 jdb-07-00018-f004:**
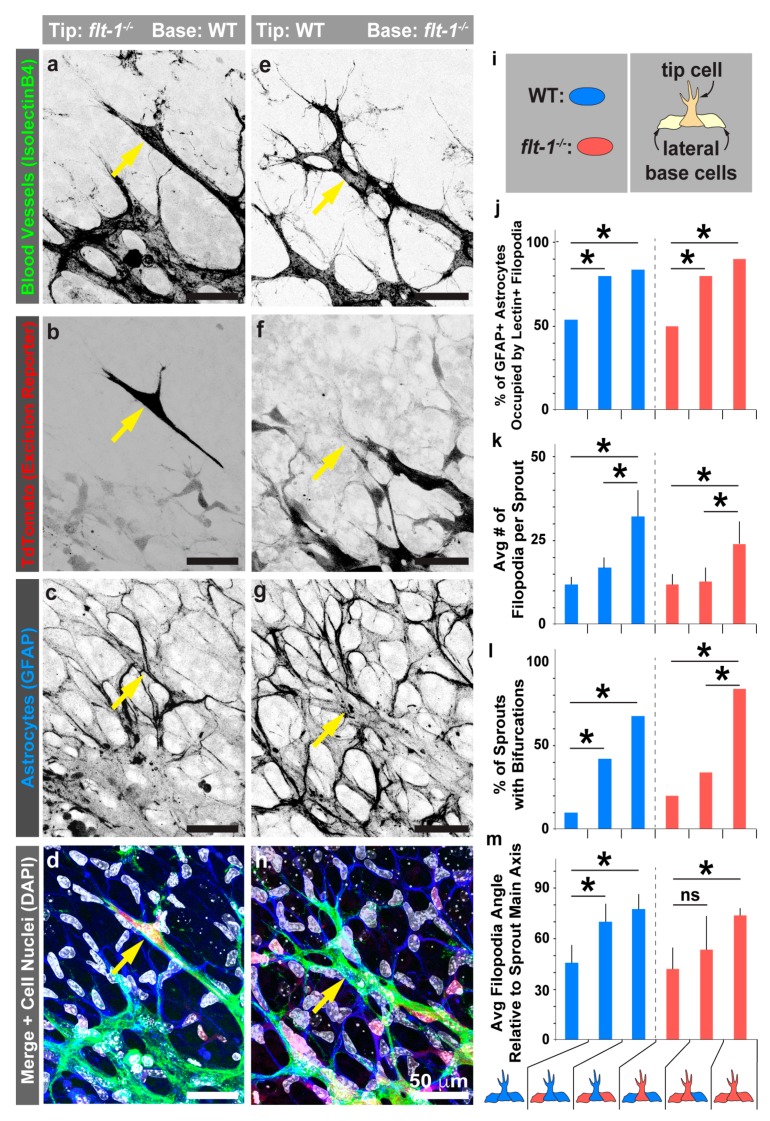
Loss of *flt-1* function in retinal lateral base cells flanking endothelial sprouts perturbs sprouting dynamics and astrocyte-guided migration independent of sprout genotype. (**a**–**h**) Representative images of postnatal day 6 (P6) mosaic conditional *flt-1* knockout (KO) mouse retinal vasculature labeled for blood vessels (isolectinB4-AlexaFluor488 in **a** and **e**, and green in merged images **d** and **h**), for *flt-1* excised cells (TdTomato reporter in **b** and **f**, and red in merged images **d** and **h**), and for astrocytes (GFAP-Alexa Fluor633 secondary in c and g, and blue in merged images **d** and **h**). Cell nuclei labeled with DAPI (white in merged images **d** and **h**). Scale bar, 50 μm. The yellow arrows indicate leading endothelial cells within sprouts emerging in mosaic areas. (**i**) Schematic for mosaic region identification and classification. (**j**–**m**) Indicated quantifications relative to mosaic sprout configuration (at bottom). (**j**) Percentage of GFAP+ astrocyte area occupied by IsolectinB4+ endothelial cell filopodia for each mosaic configuration. The blue bars represent areas in which the sprout proper was WT and the red bars represent conditional *flt-1* KO sprouts. * *p* < 0.05 by Chi square test. (**k**) Average number of filopodia per endothelial cell sprout for each mosaic configuration. The values are averages and the error bars represent standard error. * *p* < 0.05 by ordinary one-way ANOVA, followed by Tukey’s multiple comparison test. (**l**) Percentage of sprouts with bifurcated extensions for each mosaic configuration. * *p* < 0.05 by Chi square test. (**m**) Average filopodial angle relative to the sprout main axis for endothelial cell sprouts within each mosaic configuration. The values are averages and the error bars represent standard error. * *p* < 0.05 by ordinary one-way ANOVA followed by Tukey’s multiple comparison test. Measurements taken from *n* = 6 mosaic conditional *flt-1* KO mice, and each parameter measured for *n* = 6,7 tip cells from each tip cell:lateral base cell configuration (i.e., 6 configurations).

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
