# Peer review of "Blood Vessel Patterning on Retinal Astrocytes Requires Endothelial Flt-1 (VEGFR-1)"

_jdb, 2019, doi:10.3390/jdb7030018_

Round 1

Reviewer 1 Report

Chappell et al report that Flt1 is non-cell autonomously required to regulate sprouting angiogenesis in the mouse retina by regulating VEGFA availability. The authors report that sFlt1 produced by endothelial cells at the base of a growing sprout determines the subsequent behavior of the tip cell. These findings are broadly in line with previously published work from this group in other systems as well as reports from others and support the general view of how Flt1 regulates VEGFA signaling during sprouting angiogenesis. The novelty associated with this work lies in the use of mosaic analysis in the mouse retina to demonstrate the non-cell autonomous nature of the phenotype.  

Specific comments:

Line 166: I have no issue with the use of UBC-cre driver for the retina experiments, however explanation should be given for why UBC cre activity is seemingly confined to the endothelial compartment. Is it known if it is the cre or the reporter allele expression that is lower in non-endothelial cells? Is UBC-cre creating the “general excision of flt-1” stated on line 183? Perhaps clearer definintion as “UBC-cre deleted” would be more appropriate and simpler for the reader to follow given the apparent preference of this cre to delete in endothelial cells.  

Line 177-178: Care should be taken when defining the population of aSMA+ cells as "pericytes" given this is not really true. aSMA does not label all the pericytes in the vascular network, which would instead require NG2 or desmin staining. As such, the authors cannot say from the data shown that “pericytes” are negative for FLT1, only that aSMA+ mural cells are. Furthermore, there is no data co-staining for astrocytes that enables the statement that astrocytes are negative for FLT1. Finally, the examples of FLT1 negative cells shown in the aSMA co-stain image overlap with isoB4+ cells suggesting there are endothelial cells that are FLT1-ve. The authors should comment on this and refer here to other studies looking at Flt1 expression.

Figure 3: need to provide number of individual cells and animals that were included in the quantification data in panels g & h.

Figure 4: need to provide number of individual cells and animals that were included in the quantification data in panels j-m.

Author Response

Please see the attachment - one document addresses both reviewer concerns.

Reviewer 2 Report

Well written and presented manuscript from Chappell et al. showing in vivo how endothelial FLT1 shapes vascular morphogenesis on retinal astrocytes. My main concern is the limited amount of data on the novel part of the story. In fact, figure 1 and 2 are basically a reproduction of the same experiment performed by Ho et al. in 2012 (ref 32) by using a different Cre lines that still does the same job, ie universal deletion of loxP sites temporally induced by tamoxifen injection. Moreover the concept that FLt1 secretion by ECs flanking a sprout-tip is important for the prcise guidance of a new sprout (main body of fig 4) has been suggested by the same group in 2009 (Dev Cell). As I said the only novelty is how this mechanism of tight regulation of sprouting guidance relates to astrocytes. I would move fig 1 and 2 to supplementary as not novel and if possible expand the figures concerning the astrocyte-related data. 

Other than this, just a couple of suggestions for the text and 2 references that need to be replaced/discussed, see below:

row 24-26: it seems counterintuitive that defective sprouting and filopodia result in hypersprouting. I would suggest replacing "defective" with for example "altered"

row 26 and 278: replace “absent” with “in the absence of”

row 183: reference about the endothelial selective ablation of flt1 is missing

Row 184: in the retina “ECs”. ECs is missing

In fig2b it’s not clear what arrows and arrowheads are pointing to.

In fig3, the mutant picture shows a denser gfap+ network: either change representative example or quantify network density. The authors should focus on the gfap network ahead of the vascular front, because the one in the remodelled areas behind the vascular front (as currently shown and analysed in fig3) might undergo regression according to blood vessel pruning

Row 247, references are misleading, as main source of VEGFA during retinal vascularisation are INL neurons, see Okabe et al., 2014 Cell.

Fig4: the authors should validate at least once with a flt1 Ab or ISH that tdTomato expression effectively correlates with Flt1 knock down.

It’s a very nice figure and central for the authors’ take away message, but as it is, I don’t find clear where the base cells are. For example, authors should show also dapi after masking out all non endothelial dapi staining to have an idea of the number of cells involved in each sprout.

Row 280-283: I found the end of this sentence a bit convoluted. Can it be clarified?

Author Response

(The authors gave the same response as above.)
